# Rapid and Reproducible MALDI-TOF-Based Method for the Detection of Vancomycin-Resistant *Enterococcus faecium* Using Classifying Algorithms

**DOI:** 10.3390/diagnostics12020328

**Published:** 2022-01-27

**Authors:** Ana Candela, Manuel J. Arroyo, Ángela Sánchez-Molleda, Gema Méndez, Lidia Quiroga, Adrián Ruiz, Emilia Cercenado, Mercedes Marín, Patricia Muñoz, Luis Mancera, David Rodríguez-Temporal, Belén Rodríguez-Sánchez

**Affiliations:** 1Clinical Microbiology and Infectious Diseases Department, Hospital General Universitario Gregorio Marañón, 28007 Madrid, Spain; lidia.quirogam@gmail.com (L.Q.); arauko.bmcelta@gmail.com (A.R.); emilia.cercenado@salud.madrid.org (E.C.); mercedes.marinar@salud.madrid.org (M.M.); patricia.munoz.garcia@salud.madrid.org (P.M.); mbelen.rodriguez@iisgm.com (B.R.-S.); 2Instituto de Investigación Sanitaria Gregorio Marañón, 28007 Madrid, Spain; 3Clover Bioanalytical Software, Av. del Conocimiento, 41, 18016 Granada, Spain; manuel.arroyo@cloverbiosoft.com (M.J.A.); angelamolleda97@gmail.com (Á.S.-M.); gema.mendez@cloverbiosoft.com (G.M.); luis.mancera@cloverbiosoft.com (L.M.); 4CIBER de Enfermedades Respiratorias (CIBERES CB06/06/0058), 28029 Madrid, Spain; 5Medicine Department, School of Medicine, Universidad Complutense de Madrid, 28040 Madrid, Spain

**Keywords:** enterococci, vancomycin resistance, MALDI-TOF, mass spectrometry, peak analysis

## Abstract

Vancomycin-resistant *Enterococcus faecium* represents a health threat due to its ability to spread and cause outbreaks. MALDI-TOF MS has demonstrated its usefulness for *E. faecium* identification, but its implementation for antimicrobial resistance detection is still under evaluation. This study assesses the repeatability of MALDI-TOF MS for peak analysis and its performance in the discrimination of vancomycin-susceptible (VSE) from vancomycin-resistant isolates (VRE). The study was carried out on protein spectra from 178 *E. faecium* unique clinical isolates—92 VSE, 31 VanA VRE, 55 VanB VRE-, processed with Clover MS Data Analysis software. Technical and biological repeatability were assayed. Unsupervised (principal component analysis, (PCA)) and supervised algorithms (support vector machine (SVM), random forest (RF) and partial least squares–discriminant analysis (PLS-DA)) were applied. The repeatability assay was performed with 18 peaks common to VSE and VRE with intensities above 1.0% of the maximum peak intensity. It showed lower variability for normalized data and for the peaks within the 3000–9000 *m*/*z* range. It was found that 80.9%, 79.2% and 77.5% VSE vs. VRE discrimination was achieved by applying SVM, RF and PLS-DA, respectively. Correct internal differentiation of VanA from VanB VRE isolates was obtained by SVM in 86.6% cases. The implementation of MALDI-TOF MS and peak analysis could represent a rapid and effective tool for VRE screening. However, further improvements are needed to increase the accuracy of this approach.

## 1. Introduction

*Enterococcus faecium* are Gram-positive, non-spore forming, facultative anaerobic cocci that can be found as part of the microbiota of the human gastrointestinal tract [1]. Because of its genomic plasticity and its adaptation to harsh conditions, *E. faecium* has become a health threat due to its ability to rapidly spread and cause outbreaks in hospital settings [2,3]. Adding up to the intrinsic antibiotic resistance displayed by *E. faecium*, infections caused by strains with acquired resistance to certain antimicrobials are of special interest [4]. In the past years, vancomycin resistance in *E. faecium* has been a major concern owing to its rise and rapid spread of high-risk clones among hospitalized patients [5]. Therefore, *E. faecium* was included in the ESKAPE group of highly resistant microorganisms because of its ability to “escape” the action of conventional treatments [6]. Vancomycin resistance is by far the biggest threat regarding enterococci, mainly *E. faecium*, due to its position as first-line treatment for multidrug-resistant enterococcal infections [7,8]. 

Susceptibility to vancomycin can be routinely determined in the clinical microbiology laboratory using two approaches: (i) phenotypically, by the standard broth antimicrobial susceptibility testing microdilution method or by gradient diffusion, or (ii) genotypically, by amplification of the *vanA*/*vanB* genes and subsequent analysis of the specific amplicons [9,10]. The first approach has a turnaround time of approximately 2 days. Although the implementation of molecular methods provides final results in 1–3 h after isolation in culture, its cost in laboratory reagents is high.

While the usefulness of matrix-assisted laser desorption/ionization time-of-flight mass spectrometry (MALDI-TOF MS) in the microbiology laboratory for bacterial identification is settled, its implementation for antimicrobial resistance detection is not well standardized yet [11]. 

In the last years, MALDI-TOF MS has been proposed as a fast and cost-efficient method for the detection of some antimicrobial resistance mechanisms, such as β-lactamase activity [12], discrimination of methicillin-resistant *S. aureus* [13] or the detection of the *cfiA* gene in *B. fragilis* [14,15]. As for vancomycin resistance in *Enterococcus*, a few studies have been published recently but with variable results [16,17,18].

The use of MALDI-TOF MS in the clinical microbiology laboratory will be of interest as a rapid approach for the differentiation of VRE from VSE, based on their mass spectra protein profile.

The main objective of this study was the development of a MALDI-TOF-based classifying algorithm for the discrimination of vancomycin-resistant *E. faecium* (VRE) from vancomycin-susceptible *E. faecium* (VSE).

## 2. Materials and Methods

### 2.1. Bacterial Strains

A total of *n* = 178 *E. faecium* strains were included in the study and considered as the “classification set” (92 VSE, 31 VanA VRE, 55 VanB VRE). The isolates were collected consecutively throughout the years 2017 to 2019 from clinical samples of patients admitted at Hospital General Universitario Gregorio Marañón in Madrid (HGUGM) (Figure 1). Only one sample per patient was selected.

Strains were selected from blood cultures (*n* = 95) and rectal swabs (*n* = 83) (Appendix A). Clonality was clinically and epidemiologically discarded by analyzing the date, patient location and department where the inpatients were located. No clinical outbreak was detected during the period of study. All strains were isolated from inpatient clinical samples, characterized and kept frozen at −80 °C for further analysis. For this study, samples were thawed and cultured overnight at 37 °C in Columbia blood agar.

### 2.2. Antibiotic Susceptibility Testing for Vancomycin

Antimicrobial susceptibility testing was performed with the automated microdilution method Microscan^®^ System (Beckman-Coulter, Brea, CA, USA) using PM33 panels following the manufacturer’s guidelines. Vancomycin and teicoplanin breakpoints were stablished as indicated by the EUCAST (2021) v. 11. The results obtained were confirmed by real-time PCR for the amplification of the *vanA* and *vanB* genes [9]. In addition, the presence of the vancomycin resistance genes was confirmed a second time by the implementation of the commercial Xpert^®^ vanA/vanB cartridges (Cepheid, Sunnyvale, CA, USA).

### 2.3. Identification of the Isolates by MALDI-TOF MS

Bacterial strains were analyzed by MALDI-TOF MS in an MBT Smart MALDI Biotyper (Bruker Daltonics, Bremen, Germany) using the updated database containing 9957 mass spectra profiles (MSPs). A few bacterial colonies of each enterococcal isolate were spotted onto the MALDI target plate. On-plate protein extraction was performed by applying 1 µL of formic acid on each target spot and letting it dry at room temperature before adding 1 µL of HCCA matrix solution (Bruker Daltonics, Bremen, Germany), following the manufacturer’s instructions. Spectra were acquired in positive mode in the range of 2000 to 20,000 Da, applying default settings [19].

### 2.4. Spectra Acquisition and Pre-Processing

Each isolate was analyzed in two different spots from the MALDI target plate, and each spot was read twice, obtaining four spectra per strain [20]. Protein spectra were visually inspected with FlexAnalysis (Bruker Daltonics, Bremen, Germany) and aligned with the genus-specific peak at 4428 *m*/*z*, present in all isolates [16,21]. Outlier spectra and zero lines were discarded.

After the previous screening, protein spectra were processed with Clover MS Data Analysis software (https://platform.clovermsdataanalysis.com, accessed on 30 November 2021, Clover Biosoft, Granada, Spain). For classification purposes, peak matrices were generated in the range of 2000 to 20,000 *m*/*z*. For this goal, pre-processing was performed as follows: a Savitzky–Golay filter (window length 11; polynomial order 3) was applied for smoothing spectra, and then the baseline was removed by the top-hat filter method (factor 0.02).

Processed spectra were aligned using the following method: replicates within the same spot were aligned to create an average spectrum per spot (shift medium; linear mass tolerance 2000 ppm). Then, average spectra from each replicated spot were aligned, and thus one average spectrum per sample was obtained. Finally, average spectra from different isolates were aligned together.

### 2.5. Repeatability Test

A repeatability study was performed to determine the robustness of MALDI-TOF-MS-based bacterial classification. For this purpose, 20 different *E. faecium* isolates (VSE, *n* = 10; and VRE strains, *n* = 10, 5 VanA VRE and 5 VanB VRE) were randomly selected and considered as the “repeatability set” (Appendix A). For intra-spot repeatability, two spectra per spot were acquired, whilst for inter-spot repeatability (or technical repeatability), an average spectrum was built from each spot. Finally, each isolate was subcultured during three consecutive days, and their average spectra were compared in order to evaluate the biological (inter-day) repeatability of the method (Figure 2). This methodology has been described by Oberle et al., 2016 [20]. A final number of 12 spectra per isolate was obtained.

All spectra were pre-processed using Clover MS Data Analysis Software following the pipeline described in Section 2.4. The post-processed spectra (smoothed and baseline removed) were aligned (shift medium; linear mass tolerance 2000 ppm) to obtain an average spectrum for each spot. This process was repeated to obtain a single average spectrum per day for each isolate. 

Once the average Day 1 spectra from all isolates were aligned, a first assay was performed to identify all common peaks and establish their characteristic peak profiles. Peak finding was carried out by applying a threshold filter (0.01), so peaks with less than 1.0% of the maximal intensity recorded were discarded. Group-specific peaks were then searched in the repeatability set by the mass position method (constant mass tolerance 0.2 Da; linear mass tolerance 500 ppm) at the shot, spot and day levels.

The coefficient of variation (% CV) of the intensities registered for each of these common peaks was calculated from raw spectra and from spectra normalized with the TIC method in two different ways: (i) normalizing by TIC the peaks previously found in raw spectra (pTIC) and (ii) normalizing first the entire spectra by TIC and then finding the common peaks (TICp). Results from both methods and from raw data were compared for intra-spot, technical and biological repeatability. 

In addition, Pearson correlation coefficient (*p*) was applied to protein spectra from VRE and VSE isolates to measure how biological variation could affect the discrimination of these two groups and how reproducibly the discrimination could be performed. 

This peak study also included the calculation of arithmetic and post-alignment means and the subsequent comparison between them. The arithmetic means for spots and days were calculated directly from the intensity peak values of each shot. On the other hand, the post-alignment means were calculated automatically with Clover MS Data Analysis software after replicated average spectra were aligned. Both mean values were compared at spot and day levels. The assay was repeated without normalization and for both TIC methods. Thereby, the automatic alignment and replicate software process could be compared with the ideal arithmetic model in the three normalization cases. Once the Shapiro–Wilk and Levene test was applied to verify the normal distribution and the homoscedasticity of the data, a Student’s t-test was performed to verify whether the null hypothesis of equality of means was fulfilled for the three normalization methods. 

### 2.6. Classification of E. faecium Isolates Based on Their MALDI-TOF MS Protein Spectra

Protein spectra from *E. faecium* isolates were acquired as described in Section 2.4. Clover MS Data Analysis Software was applied to differentiate (1) susceptible from resistant *E. faecium* strains and (2) isolates hosting *vanA* and *vanB* resistance genes. For this purpose, three different methods were evaluated: (i) a “full-spectrum method”, in which the matrix obtained included all peak intensities from the spectrum separated by 0.5 Da regardless of their intensity. Their intensities were then normalized by total ion current (TIC) normalization. The other two methods used a peak matrix generated by a “threshold method”, in which only peaks with intensities above 1.0% of the maximum peak intensity (0.01 factor) were chosen. The difference in the last two methods was the order in which TIC normalization was applied: (ii) before (TICp) or (iii) after (pTIC) searching peaks by the threshold method.

The three peak matrices described above were used as input data for three different supervised machine learning algorithms: partial least squares–discriminant analysis (PLS-DA), support vector machine (SVM) and random forest (RF). These algorithms were first tested for the discrimination VSE and VRE isolates and, secondly, for the differentiation of VRE isolates hosting *vanA* and *vanB* genes.

Internal validation of the results provided by each algorithm was assayed using k-fold cross validation (*k* = 10) as previously described [22]. Briefly, data were randomly split into 10 data subsets of the same size. The algorithms were trained with nine of them and the remaining subset was used as a test set for internal validation. This process was iterated 10 times (once for each of the 10 subsets) and the accuracy rate of the classification was recorded [23].

Furthermore, a search for potential biomarkers was performed to use group-specific protein peaks as markers for the correct classification of VanA and VanB VRE and VSE strains. For this purpose, 178 average spectra (one per sample) from Day 1 were considered as input data for the Biomarker Analysis application within Clover MS Data Analysis software. The threshold method was applied as explained above, and peaks were merged with 0.5 Da and 300 ppm as constant and linear mass tolerance, respectively.

## 3. Results

### 3.1. Identification by MALDI-TOF MS

All isolates (*n* = 178) included in this study were correctly identified as *E. faecium* by MALDI-TOF MS with score ≥ 2.0. Identifications for this microorganism were consistent along the top 10 identifications provided by MALDI-TOF MS since this is a common pathogen well represented in the updated commercial library.

### 3.2. Repeatability Study

When the different methods for peak finding described before were applied to the Day 1 spectra from the “classification set”, a total of 18 common peaks were found in all *E. faecium* (VSE and VRE) protein spectra with intensities above 1.0% of the maximum peak intensity (Table 1). The CV means of the intensities from these 18 common peaks were compared with the peaks present in the spectra from the repeatability set at intra-spot, inter-spot and inter-day levels.

Comparing the means of the CVs in the repeatability set showed lower CV values at the intra-spot level than at the inter-spot and inter-day levels. For non-normalized raw data, the average CVs were CV_intra-spot_ = 15.35, CV_inter-spot_ = 29.29 and CV_inter-day_ = 31.25. The same pattern was shown for the TICp (CV_intra-spot_ = 8.46, CV_inter-spot_ = 20.88 and CV_inter-day_ = 20.66) and pTIC (CV_intra-spot_ = 7.99, CV_inter-spot_ = 19.91 and CV_inter-day_ = 19.30) methods (Figure 3). Additionally, data normalization allowed the reduction of CV values at the three levels (*p* < 0.0001). Differences between both normalization methods (TICp and pTIC) varied between 0.47 for CV_intra-spot_ and 1.36 for CV_inter-day_, demonstrating that both allowed for reduced CV values at intra-spot (44.80% and 47.94% for TICp and pTIC, respectively), inter-spot (28.71% and 32.0%) and inter-day (33.89% and 38.24%) levels. Reduced CV values were consistently recorded for VSE and VRE isolates alike (Appendix A).

The variability of the 18 common peaks found in all *E. faecium* isolates showed that CV values were lower for peaks between 3000 and 9000 *m*/*z* at intra-spot (Figure 4A), inter-spot (Figure 4B) and inter-day levels (Figure 4C), especially when data were normalized, showing different means (*p* < 0.05) versus the 2000 and 3000 *m*/*z* range (Appendix A). The 5974.6 *m*/*z* peak showed higher CV values at the three levels but the region between 6000 and 9000 *m*/*z* showed lower variation again, although the CVs were higher for inter-spot and inter-day repeatability (Figure 4B,C). These results support the fact that the central *m*/*z* region of the spectrum is the most reliable for peak analysis [24].

The Pearson correlation coefficient was applied to inter-day repeatability, showing a mean of 0.94 factor for all samples (Appendix A). This value was higher for VRE (*p* = 0.95) than for VSE isolates (0.93). Within the VRE group, the *p* value for isolates hosting the *vanA* mechanism was 0.98 versus 0.92 for the isolates with the *vanB* resistance gene.

Arithmetic mean, as a representative value of peak intensity, and post-alignment mean were calculated and compared (Table 2). For non-normalized data, the arithmetic mean was higher than the post-alignment mean for inter-spot and inter-day levels, unlike the case for the TICp normalization method. Regarding pTIC normalization, the arithmetic and post-alignment means were identical. Furthermore, the homoscedasticity of the data and their normal distribution were checked by the implementation of the Levene and Shapiro–Wilk tests respectively. The t-Student test was then performed to check whether the null hypothesis of equality of means could be accepted among arithmetic and post-alignment means. In all possible comparisons, *p*-values obtained were >0.05 (Appendix A), showing that, even when a post-alignment was performed, the intensity values were not affected. The fact that both means in all methods did not show statistical differences, proved the high repeatability of the assays based on protein spectra analysis.

### 3.3. Classification of the Isolates Using Machine Learning

Three peak matrices generated (full-spectrum method and threshold methods TICp and pTIC) with data from the “classification set” were used as input data to test the capacity of the algorithms to discriminate VRE (VanA + VanB) from VSE isolates (Appendix A). The 10-fold cross validation results for the SVM algorithm showed the best accuracy with 80.9% and an F1 score (the harmonic mean of the sensitivity and the accuracy of the model) of 80.5% for the full-spectrum-TIC method (Table 3A).

The same procedures were also applied for the discrimination of VanA from VanB VRE strains (Appendix A). In this approach, PLS-DA algorithm with TICp method provided 86.65% correct classification (Figure 5). This algorithm achieved 89.09% of predictive value for identifying VanB VRE strains in a 10-fold cross validation (Table 3B).

The biomarker analysis revealed the presence of two potential resistance biomarkers at 6891.33 and 5095.01 *m*/*z* (Table 4). These two peaks showed AUC values greater than 0.8 in its receiver operating characteristic (ROC) curve (Appendix A). The 5095.01 *m*/*z* peak allowed the discrimination of the VRE strains from the VSE strains (Appendix A). This peak was present in 82 of the 86 VRE strains in this study. Furthermore, the AUC for peak at 6891.33 *m*/*z* allowed the discrimination of VRE isolates hosting the *vanA* resistance mechanism from those carrying the *vanB* one. Its intensity was higher in all VanA *E. faecium* isolates tested with an AUC of 0.831 and a CV value of 22.99% (Appendix A). These results were obtained by applying a threshold (0.01 factor) after a TIC normalization using the 178 pre-processed samples of the classification set.

## 4. Discussion

The application of MALDI-TOF MS coupled with data analysis was shown to be a reproducible methodology—CV values ≤20.88 for normalized data—that allowed the discrimination of VRE (VanA + VanB) from VSE in 80.9% of the cases using SVM and the correct differentiation of 86.6% of the *E. faecium* VanB isolates from the VanA VRE isolates by PLS-DA. Specific peaks for the discrimination of the studied isolates were found: the 5095.01 *m*/*z* peak was present in 82/86 VREs. Although this peak has already been described as a biomarker for VRE isolates [25], its real meaning is currently under debate. Brackmann et al. (2020) recently reported the sequencing of the 5095.01 *m*/*z* protein and identified the protein hiracin, a secretory protein encoded by the hirJM79 gene, whose role in the vancomycin resistance mechanism remains unknown [26]. Although our study supports the value of the 5095.01 *m*/*z* peak as a marker for VRE isolates, caution should be exercised when using this peak for VRE presence until further studies unravel its correlation with vancomycin resistance.

The 6891.33 *m*/*z* peak was found as a biomarker for VanA VRE isolates. Both the AUC (0.831) and the CV values (22.99%) indicate the uniqueness and repeatability of this marker. Although this peak had been related before with different clonal complexes and sequence types [17], its correlation with VanA VRE isolates had not been reported so far [27].

The repeatability study carried out with 20 *E. faecium* isolates (10 VSE, 5 VanA VRE and 5 VanB VRE) demonstrated that the lowest CV values for peak intensities were obtained for normalized data (*p* < 0.0001), regardless of the order in which normalization and peak finding is performed. Intra-spot repeatability showed the highest rate of repeatability (CV values ranging from 7.99 to 8.46), although the CV values for inter-spot and inter-day repeatability ranged between 19.91 and 20.88 for inter-spot variability and between 19.30 to 20.66 for inter-day repeatability. Similar CV values (6.5–17%) have been reported in a study that evaluated the technical repeatability of MALDI-TOF MS for quantitative protein profiling [28]. Therefore, this methodology is considered reproducible and feasible for peak analysis, especially in the range of 3000–9000 *m*/*z*. In addition, 13 of the 18 most representative common peaks for all *E. faecium* and also both biomarker peaks for VRE vs. VSE and VanA VRE vs. VanB VRE differentiation are located within this spectrum range. The implementation of the methodology described in this study could provide standardization for data comparison with other studies analyzing antimicrobial resistance with MALDI-TOF MS.

Previous studies have shown the ability of MALDI-TOF MS to differentiate among *E. faecium* vancomycin-resistant high-risk clones, clonal complexes and sequence types with different success rates [16,17,18]. Differentiation between VRE and VSE isolates has also been reported. Griffin et al. reported 88.45% correct discrimination using SVM and 88.24% with the implementation of the genetic algorithm [25]. This algorithm also allowed a discrimination rate of 92.4% for VanA VRE from VSE isolates in the study developed by Nakano et al. [27]. In our case, the application of SVM, RF and PLS-DA algorithms provided 80.9%, 79.2% and 77.5% correct classification of VRE and VSE isolates using the full-spectrum method. Moreover, discrimination between VanA and VanB VRE isolates was achieved in 86.6% of the cases by applying the TICp method. Discrepancies in the peaks used for discrimination of the different *E. faecium* groups were detected with the two previous studies: none of the peaks included in the study by Nakano et al. were found relevant in our models; besides, only the 5095.01 *m*/*z* peak (5094.7 *m*/*z* in Griffin et al.) was common to both studies for the discrimination of VRE from VSE. In our predictive models, the 6603 *m*/*z* peak proposed by Griffin et al. for the discrimination of VanA from VanB VRE was not considered discriminative. Instead, the 6891.78 *m*/*z* peak served this purpose in our study.

Although further studies are requested in order to clarify the role of the 5095.01 *m*/*z* peak for the routine detection of VanB VRE isolates, we propose the detection of this peak in combination with the 6891.78 *m*/*z* peak for the detection of suspected VanA VRE, for the differentiation of VRE using MALDI-TOF MS. Despite VanB VRE isolates being more prevalent in our setting, VanA VRE isolates have been reportedly correlated with hospital infections in different European countries [8]. Thus, both biomarker peaks could be used for rapid screening of VRE isolates with MALDI-TOF MS.

One of the limitations of this study is the lack of genomic background for the analyzed isolates. Only the genes encoding vancomycin resistance were targeted, and its presence or absence was confirmed by molecular methods. Although this information was useful for the development of predictive models, more comprehensive information about our VRE isolates could help obtain higher discrimination power from the applied algorithms and also explain the misclassifications from the current models.

Another limitation of the study is that clonality of the isolates was only analyzed within a clinical and epidemiological approach. HGUGM is a tertiary hospital where clinical departments are in different wards, far from each other, and contact among them is sporadic—as each department has its own medical staff—so an outbreak affecting different departments is unlikely. The theory of non-clonality is also supported by Griffin et al., since the 5095.01 *m*/*z* peak was also found in their study, performed in a different continent [25]. We acknowledge that clonality cannot be fully discarded as we lack a deep whole-genome sequencing approach. More accurate molecular methods are needed to rule out that the *E. faecium* isolates analyzed in this study belong to the same clone. However, our results support the importance of a previously described peak for differentiating VRE from VSE isolates and add a new specific biomarker for the discrimination of VanA VRE strains.

The classification accuracy of the applied algorithms has shown to be <90% in all cases. Therefore, further studies with well-characterized isolates sourced from different geographic origins are needed to confirm the results obtained in this work and improve them if possible.

## 5. Conclusions

In conclusion, MALDI-TOF MS has demonstrated acceptable discrimination of *E. faecium* isolates beyond species assignment. Although further refining is requested and isolates from different clones and origins have to be included in predictive models in order to understand how they are discriminated by MALDI-TOF MS, protein profiling could become a suitable tool for the rapid detection of VRE in clinical microbiology laboratories. Its implementation could be key for the control of VRE isolates in hospital settings.

## Figures and Tables

**Figure 1 diagnostics-12-00328-f001:**
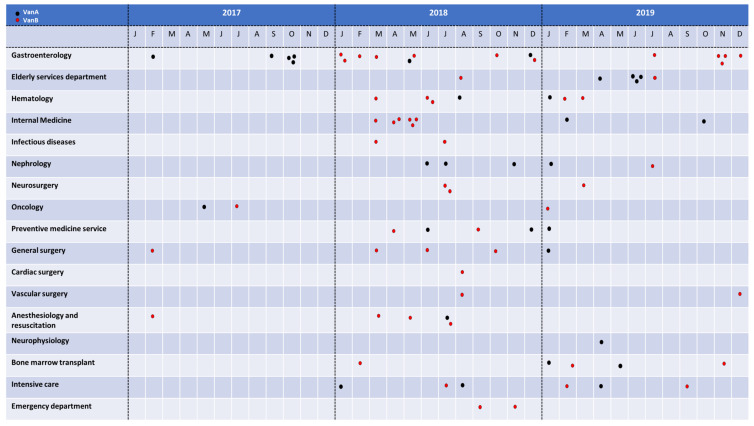
Distribution of the *E. faecium* isolates analyzed in this study by hospital departments and date. VanA *E. faecium* isolates are shown with black dots and VanB isolates with red dots.

**Figure 2 diagnostics-12-00328-f002:**
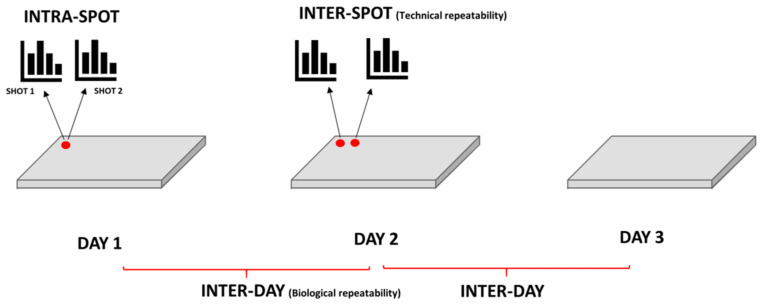
Graphic representation of 12 replicates spectra for each isolate: 2 spectra are acquired per spot; 2 spots analyzed each day and compared for 3 consecutive days.

**Figure 3 diagnostics-12-00328-f003:**
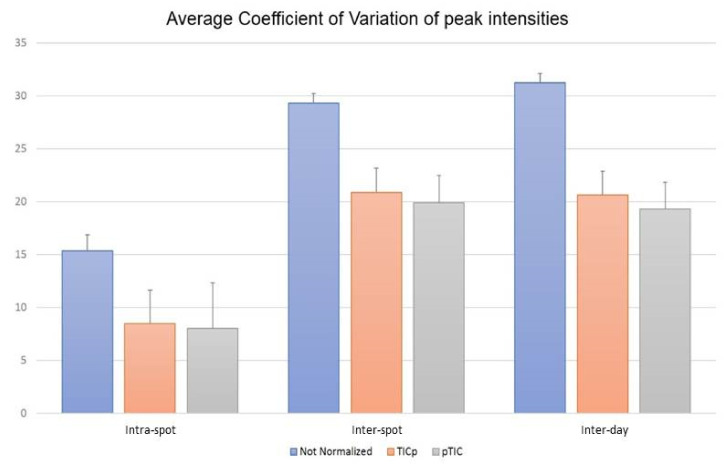
Coefficient of variation (CV) means for the intensity of the 18 common protein peaks of *E. faecium* analyzed at the intra-spot, inter-spot and inter-day levels using raw data (non-normalized) and both normalizations methods—before (TICp) and after (pTIC) finding peaks.

**Figure 4 diagnostics-12-00328-f004:**
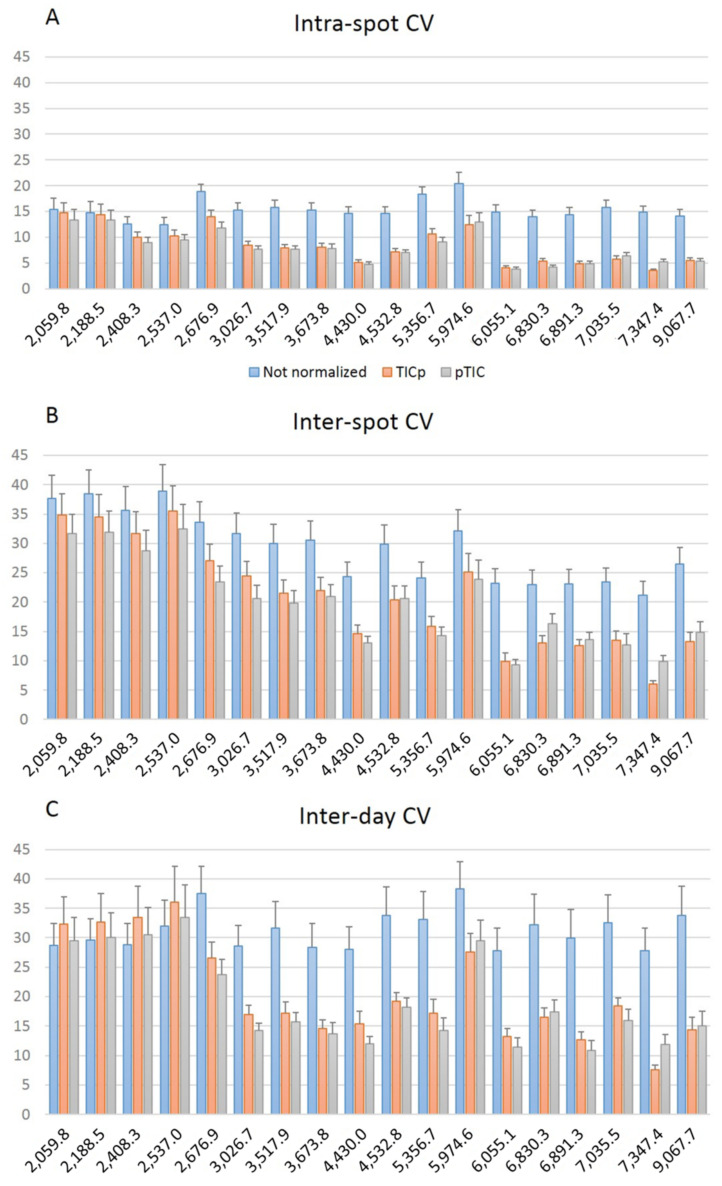
Comparison of the average coefficient of variation (CV) for the peak intensity of the 18 common peaks analyzed at intra-spot (**A**), inter-spot (**B**) and inter-day (**C**) levels.

**Figure 5 diagnostics-12-00328-f005:**
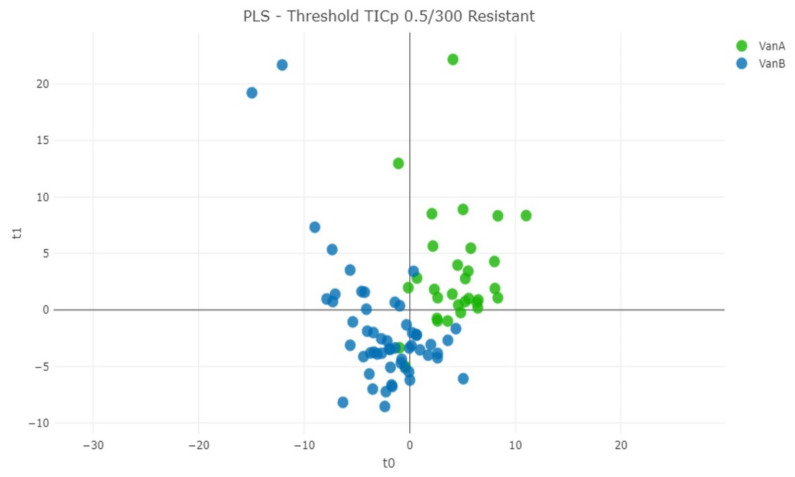
Distance plot of partial least squares–discrimination analysis (PLS-DA) machine learning algorithm for the discrimination of VanA from VanB VRE isolates using the TICp method.

**Table 1 diagnostics-12-00328-t001:** List of common peaks (*n* = 18) found in the average spectra of the vancomycin-resistant *E. faecium* and vancomycin-susceptible *E. faecium* isolates included in the classification set. CV = coefficient of variation of intensity. Mean in intensity units.

Mass (*m*/*z*)	Measurements		
	Appearance (%)	CV (%)	Mean
2059.79	178/178 (100)	53.42	4361.20
2188.53	178/178 (100)	56.61	5269.55
2408.35	178/178 (100)	49.04	3316.45
2537.05	178/178 (100)	53.06	3527.45
2676.89	178/178 (100)	51.89	2170.47
3026.71	178/178 (100)	47.65	1270.16
3517.85	178/178 (100)	56.14	1327.19
3673.77	178/178 (100)	50.43	3344.71
4430.01	178/178 (100)	46.26	17,437.72
4532.78	178/178 (100)	57.94	2296.61
5356.72	178/178 (100)	56.15	8719.83
5974.65	178/178 (100)	68.28	2531.38
6055.07	178/178 (100)	50.30	4187.74
6830.35	178/178 (100)	71.56	3020.35
6891.33	178/178 (100)	59.00	5252.80
7035.47	178/178 (100)	65.96	2656.59
7347.39	178/178 (100)	55.31	10,326.29
9062.75	178/178 (100)	66.94	2820.45

**Table 2 diagnostics-12-00328-t002:** Arithmetic and post-alignment means at spot- and day-level without normalization and with the two different normalization methods applied—before (TICp) and after (pTIC) finding peaks.

	Not Normalized		TICp		pTIC	
	Arithmetic	Post-Alignment	Arithmetic	Post-Alignment	Arithmetic	Post-Alignment
Inter-spot mean	3825.44	3792.14	0.0005	0.0005	0.056	0.056
Inter-day mean	3974.07	3955.49	0.0005	0.0005	0.056	0.056

**Table 3 diagnostics-12-00328-t003:** Discrimination of vancomycin-susceptible *E. faecium* (VSE) from vancomycin-resistant *E. faecium* (VRE) isolates and, within the latter group, differentiation between the strains hosting the *vanA* and *vanB* resistance genes. The actual classification is shown in columns and the predictive classification in rows. (**A**) Results from the support vector machine (SVM) algorithm using the full-spectrum-TIC method. Accuracy: 80.9%; F1 score: 80.46%; sensitivity: 81.4% specificity: 80.43%; positive predictive value (PPV) or precision: 79.55%; negative predictive value (NPV): 82.22%. (**B**) Scores from the partial least squares–discrimination analysis (PLS-DA) algorithm using the total ion current normalization after peak finding (TICp) method. Accuracy: 86.05%; predictive value for VanA VRE: 80.65%; predictive value for VanB VRE: 89.09%.

Actual Classification	Predicted Classification	
(A) SVM Full Spectrum	VRE	VSE
VRE	70	16
VSE	18	74
(B) PLS Threshold TICp	VanA	VanB
VanA	25	6
VanB	6	49

**Table 4 diagnostics-12-00328-t004:** Biomarker peaks for the detection of the vancomycin-resistant *E. faecium* (VRE) isolates and the discrimination of strains hosting the VanA mechanism.

Peak *m*/*z*	AUC (≥0.8)	Appearance (Total Samples)	Positive Category	Coefficient of Variation (CV)
5095.01	0.814	123/178	82/86 (resistant)	61.63%

## Data Availability

All data generated or analyzed during this study are included in this published article (and its Appendix A). Classification algorithms are available at https://platform.clovermsdataanalysis.com/ accessed on 20 December 2021.

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
