# Peer review of "Rapid and Reproducible MALDI-TOF-Based Method for the Detection of Vancomycin-Resistant Enterococcus faecium Using Classifying Algorithms"

_diagnostics, 2022, doi:10.3390/diagnostics12020328_

Round 1

Reviewer 1 Report

This was a well written paper with good scientific design. And it is recommended for publication with minor editorial changes.  They are as follows;  

  1. Figure 1 needs replacement of yellow dots with solid black so they are easily identifiable to reader.  The yellow dots don't show up.
  2. A sentence is needed to relate the 18 different species of Enterococcus faecalis to the 178 E. faecium unique clinical isolates first introduced in the abstract. 
  3. In the figure and table legends, all abbreviations must be spelled out since each must be able to stand alone without the text.  For instance, Table 3, ie., SVM and PLS, were described and defined in the text, but not in the table legend. 
  4. In figure 3, three colors are shown, but not defined.  Also, there is a typo , ie., "shots" should be "spots". 
  5.  When reading the text, there was confusion about the word "reproducibility" (example in line 323) versus "repeatability".  According to Patil et. al. 2016, the difference between repeatability and reproducibility, is as follows:  'Repeatability measures the variation in measurements taken by a single instrument or person under the same conditions, while reproducibility measures whether an entire study or experiment can be reproduced in it's entirety."  Please have an editor check out these words. 

Author Response

Point 1: Figure 1 needs replacement of yellow dots with solid black so they are easily identifiable to reader. The yellow dots don't show up.

Response: As recommended by the reviewer, yellow colour dots of Figure 1 have been replaced by solid black dots and red dots have been made “more visible” by reducing thickness of the black outline.

Point 2: A sentence is needed to relate the 18 different species of Enterococcus faecalis to the 178 E. faecium unique clinical isolates first introduced in the abstract.

Response: We do not fully understand what reviewer #1 means in this point. 178 Enterococcus faecium isolates were analyzed in this study but Enterococcus faecalis were not included. We assumed the comment is about the 18 common protein peaks used for reproducibility purposes and, therefore, we added a sentence about it in the abstract: “The reproducibility assay was performed with 18 peaks common to VSE and VRE with intensities above 1.0% of the maximum peak intensity.” Lines 25-26

Point 3: In the figure and table legends, all abbreviations must be spelled out since each must be able to stand alone without the text. For instance, Table 3, ie., SVM and PLS, were described and defined in the text, but not in the table legend.

Response: Now, all abbreviations have been spelled out in the new version of the manuscript: in Tables 1- 4 and Figures 3-5.

Point 4: In figure 3, three colors are shown, but not defined. Also, there is a typo, ie., "shots" should be "spots".

Response: As recommended by the reviewer, colors of Figure 3 have been defined at the bottom of the figure and “shots” replaced by “intra-spot”, as in the figure legend.

Point 5: When reading the text, there was confusion about the word "reproducibility" (example in line 323) versus "repeatability". According to Patil et. al. 2016, the difference between repeatability and reproducibility, is as follows: 'Repeatability measures the variation in measurements taken by a single instrument or person under the same conditions, while reproducibility measures whether an entire study or experiment can be reproduced in it's entirety." Please have an editor check out these words.

Response: The word reproducibility has been replaced by repeatability, as suggested by the reviewer.

Reviewer 2 Report

The authors report rapid method to distinguish VRE and VSE using MALDI-TOF-MS data. Classification using MALDI data is promising and various machine learning algorithms have been adapted. Generally, the developed method seems to be useful and manuscript was carefully prepared. However, this reviewer have a few comments which should be clarified by the authors.

  1. Previous researches about VRE and VSE discrimination described from line 342 should be included in introduction in order to emphasize the novelty of this work.
  2. The authors used a few different algorithms and just described the results However, there is no explanation or guidance to choose the algorithms
  3. Exact full nomenclatures of VRE and VSE are missing.

Author Response

Point 1: Previous researches about VRE and VSE discrimination described from line 342 should be included in introduction in order to emphasize the novelty of this work.

Response: As recommended by the reviewer, the previous researches about VRE and VSE discrimination have been included at the end of introduction section. “As for Vancomycin resistance in Enterococcus, a few studies have been published recently, but with variable results [16, 17, 18]” Lines 63-64.

Point 2: The authors used a few different algorithms and just described the results However, there is no explanation or guidance to choose the algorithms

Response: This area of knowledge is still under development (doi: 10.1016/j.cmi.2020.03.014). So far, different algorithms (both supervised and unsupervised) have been applied empirically to MALDI-TOF data sets. So far, different authors have implemented available algorithms and reported the rate of successful classification with them. Lately, Bayesian models have been applied to the discrimination of carbapenemase producing Klebsiella pneumoniae (https://www.biorxiv.org/content/10.1101/2021.10.04.463058v2). The good results yielded by these models predict a wider implementation of this novel approach in future research studies.

Point 3: Exact full nomenclatures of VRE and VSE are missing.

Response: Full nomenclatures of VRE and VSE are now described at the end of introduction section: “The main objective of this study was the development of a MALDI-TOF-based classifying algorithm for the discrimination of Vancomycin-resistant (VRE) from Vancomycin-susceptible E. faecium (VSE).” Lines 68-70